# The Role of LIM Kinases during Development: A Lens to Get a Glimpse of Their Implication in Pathologies

**DOI:** 10.3390/cells11030403

**Published:** 2022-01-25

**Authors:** Anne-Sophie Ribba, Sandrine Fraboulet, Karin Sadoul, Laurence Lafanechère

**Affiliations:** Institute for Advanced Biosciences, Team Cytoskeletal Dynamics and Nuclear Functions, INSERM U1209, CNRS UMR5309, Université Grenoble Alpes, 38000 Grenoble, France; anne-sophie.ribba@univ-grenoble-alpes.fr (A.-S.R.); sandrine.fraboulet@univ-grenoble-alpes.fr (S.F.); karin.sadoul@univ-grenoble-alpes.fr (K.S.)

**Keywords:** LIM kinase, development, cofilin

## Abstract

The organization of cell populations within animal tissues is essential for the morphogenesis of organs during development. Cells recognize three-dimensional positions with respect to the whole organism and regulate their cell shape, motility, migration, polarization, growth, differentiation, gene expression and cell death according to extracellular signals. Remodeling of the actin filaments is essential to achieve these cell morphological changes. Cofilin is an important binding protein for these filaments; it increases their elasticity in terms of flexion and torsion and also severs them. The activity of cofilin is spatiotemporally inhibited via phosphorylation by the LIM domain kinases 1 and 2 (LIMK1 and LIMK2). Phylogenetic analysis indicates that the phospho-regulation of cofilin has evolved as a mechanism controlling the reorganization of the actin cytoskeleton during complex multicellular processes, such as those that occur during embryogenesis. In this context, the main objective of this review is to provide an update of the respective role of each of the LIM kinases during embryonic development.

## 1. Introduction

LIM kinase 1 (LIMK1) first appeared on the scientific scene in 1994, thanks to the work of a Japanese [1] and an Australian team [2]. The following year, the LIM kinase family was extended with the description of LIM kinase 2 (LIMK2), which has an overall sequence and a domain structure similar to that of LIMK1, but the overall identity is 50–51% at the amino acid level [3]. As indicated by their denomination, they contain two LIM domains. The name LIM is an acronym of the three genes in which such a domain was first identified (LIN-11, Isl-1 and MEC-3). LIM domains are tandem zinc-finger structures that function as modular protein-binding interfaces [4]. The LIM domains of LIMKs are positioned in the amino-terminal part of the protein. These domains are followed by a central PDZ domain, a proline/serine (P/S)-rich region, and a carboxyterminal kinase domain. Thus, LIMKs belong to the PDZ-LIM protein family with which they share the common trait of influencing the actin cytoskeleton [5]. In addition, the PDZ-LIM family of proteins has been shown to mediate signals between the nucleus and the cytoskeleton, with a significant impact on organ development [6].

In 1998, the physiological substrate of LIMK1 was identified; in two “back-to-back” *Nature* publications, Arber and collaborators [7] and Yang and collaborators [8] provided evidence that LIMK1 phosphorylates cofilin. The following year, it was shown that cofilin was also the substrate of LIMK2 [9]. While cofilin remains the best-characterized substrate of LIM kinases (LIMKs), other substrates have been identified, extending the field of action of this family of kinases [10,11,12].

Cofilin is a protein that plays an essential role in the regulation of actin dynamics. It binds to actin filaments, which increases their elasticity in terms of both flexion and torsion, and severs them, leading to their depolymerization [13,14]. Cofilin is regulated by phosphorylation of the serine residue at position 3, which inhibits its actin-binding and depolymerization activities. Besides LIMKs, cofilin is phosphorylated and inactivated by testicular protein kinases (TESKs) [15,16], Nck-interacting kinase-related kinases (NRK) [17] and by Aurora kinase [18]. Cofilin is reverted to its basal unphosphorylated active state by the phosphatases slingshot 1 (SSH) [19] and chronophin [20].

From a phylogenetic point of view, gene orthologs of LIMKs are present in vertebrates, as well as in *Drosophila* and *Anopheles*, but are not found in yeast, *Caenorhabditis elegans*, *Dictyostelium*, or in plants [6,21]. Therefore, it has been proposed by Kazumasa Ohashi in his brilliant review on the role of cofilin in development, that “cofilin phosphoregulation may have evolved as a mechanism for reorganizing the actin cytoskeleton during complex multicellular processes in some higher organisms” [22].

The morphogenesis of organs and tissues during development involves the controlled arrangement of multicellular processes. Depending on the extracellular signals they receive, cells adapt their shape, motility and migration, polarization, growth, differentiation, and cell death. Because of their ability to control actin remodeling, LIMKs have a central role in many of these processes. The prominent role of LIMK1 during development is evidenced by some of the features observed in Williams–Beuren syndrome. This rare genetic disorder, first described by New Zealand’s Dr. John Cyprian Phipps Williams in 1961 and in the following year by Germany’s Dr. Alois J. Beuren, is caused by the hemizygous deletion of approximately 1.5–1.8 mega base pairs on the long arm of chromosome 7 (7q11.23), encompassing 27 genes including elastin, CLIP115 and LIMK1. Some of the clinical characteristics of Williams–Beuren syndrome [23], such as distinctive craniofacial abnormalities (broad forehead, wide mouth, full cheeks and lips, and oval ears), impaired visuospatial constructive cognition, and mild mental retardation have been specifically associated with the deletion of LIMK1 [24,25,26].

In this review, we will focus on the identified roles of LIMKs in specific developmental processes and on the LIMK-regulated signaling pathways, whose activation is central during development. In addition to descriptions of the tissue expression of LIMKs, we will base our review on studies of animal models with loss- or gain-of-function mutations in LIMKs, as well as on studies where LIMK functions are probed with specific inhibitors of LIMK activity.

Understanding the role of LIMKs during development may indeed shed light on the LIMK-dependent pathological perturbations observed not only in Williams–Beuren syndrome but also in other pathologies (autism, fragile X syndrome (see the recent review of Ben Zablah and collaborators [25]) or carcinogenesis (see reviews of Lee and collaborators [27] or of Fabrizio Manetti [28,29])).

## 2. Expression of LIMKs during Development 

The expression patterns of LIMK1 and LIMK2 during mammalian early development have been explored only recently, in a study investigating their role during porcine embryonic development. Using first immunofluorescence staining and then validation of LIMK1/2 expression by quantitative real-time polymerase chain reaction (qRT-PCR), it was found that both LIMK1 and LIMK2 are expressed at an early point in development and play a role during the cleavage and morula stages of embryos. Inactivation of LIMKs through silencing with dsRNA or the use of the LIMK inhibitor LIMKi3, which shows a good selectivity within the kinome (see the recent review by Chatterjjee and collaborators [30]), induced the disruption of adherent junctions, resulting in the early termination of development before the blastocyst stage [31]. A similar observation was made in an analysis of early mouse embryogenesis: the use of the LIMKi3 inhibitor caused the failure of early blastomere cleavage, compaction and blastocyst formation [32]. These results point to a crucial role of LIMKs during the early stages of development. They also indicate that LIMK activity is compensated for by the action of other kinases in LIMK knock-out mice, as they give birth to viable pups.

During the later stages of development (Table 1), LIMK1 is highly expressed in neuronal tissues and is present in many epithelial tissues, where its expression pattern is spatially and temporally regulated. LIMK1 is detected in embryonic skin epidermis, the heart, lung, and kidney in varying amounts, depending on the stage of development. It has been observed that LIMK1 is also found in specific cell types that undergo transitions between epithelial and mesenchymal states [33]. 

Similar to LIMK1, the LIMK2 expression pattern is also dynamically regulated in space and time during all developmental stages, and the expression of both LIMKs may overlap in some embryonic tissues [3,34,35,36,37,38]. Although also present in neuronal tissues, LIMK2 seems to be more abundantly detected in epithelia such as the digestive tract [36] (Table 1). A clear comprehension of the expression pattern of LIMKs is, however, complicated due to the existence of several splice variants [39,40,41].

**Table 1 cells-11-00403-t001:** Expression of LIMKs during development.

Publications	LIMK Isoform Species	Embryonic/Adult Tissues—Cell Lines	Experimental Procedures	Main Observations
Mizuno et al. *Oncogene*, 1994 [1]	LIMK human, rat	adult rat brain, epithelial and hematopoietic cell lines	Northern blot	High level in the rat brain. Expressed in human epithelial and hematopoietic cell lines
Bernard et al. *Cell Growth and Differentiation*, 1994 [2]	LIMK human, mouse	E13, E14, E15, E16, E18, P0 mouse brain, adult human brain, mouse brain, heart, liver, muscle mouse olfactory epithelial cell lines	Northern blot, RNase protection assay in situ hybridization, immunohistochemistry	Identification of LIMK. Expressed in human and mouse brain and olfactory epithelial cell lines
Ohashi et al. *Journal of Biochemistry*, 1994 [42]	LIMK chicken	adult lung, brain, kidney, liver, gizzard, intestine, spleen	Northern blot	Expressed in lung, brain, kidney, liver, spleen, gizzard and intestine
Cheng and Robertson *Mechanisms of Development*, 1995 [43]	LIMK mouse	E8.5, E11.5, E15.5 brain, olfactory system, gut, trophoblast giant cells adult brain, ovary, testis, skin, lung	Northern blot, in situ hybridization	Variable expression rates depending on the stage of development and the tissue
Okano et al. *Journal of Biological Chemistry*, 1995 [34]	LIMK1 LIMK2 human	adult brain, skeletal muscle, heart, placenta	Northern blot	LIMK1 expressed in all tissues, with highest amounts in the brain. Two LIMK2 isoforms: longer in all tissues, smaller only in skeletal muscle and heart
Pröschel et al. *Oncogene*, 1995 [44]	LIMK1 mouse	adult spinal cord, brain, cranial nerve, dorsal root ganglia	Northern blot, in situ hybridization	Nervous system expression of LIMK1
Nunoue et al. *Oncogene*, 1995 [3]	LIMK1 LIMK2 rat	adult brain, various tissues	Northern blot	LIMK1 in the brain, LIMK2 in various tissues
Ikebe et al. *Genomics*, 1997 [45]	LIMK2 mouse	adult brain, thymus, lung, heart, stomach, spleen, kidney, intestine, liver, testis	RT-PCR	LIMK2a and LIMK2b isoforms expressed in various tissues
Koshimizu et al. *Biochemistry and Biophysical Research Communications*, 1997 [39]	LIMK2 mouse	E10 to E18 embryos adult brain, heart, lung, spleen, thymus, kidney, stomach	Northern blot	LIMK2a and LIMK2b isoforms expressed in various tissues
Mori et al. *Molecular Brain Research*, 1997 [36]	LIMK1 LIMK2 rat	E12, E14, E16, E18 embryo	In situ hybridization	LIMK1 and LIMK2 expressed in brain. Differential expression of LIMK1 and LIMK2 in epithelia. High expression in extra-embryonic tissues
Takahashi et al. *Developmental Dynamics*, 1997 [46]	LIMK1 xenopus	Stage 2 to 40 cleavage, gastrula, blastula, neurula	Northern blot, in situ hybridization	Variable expression rates during development. Important role of XLIMK1 in neural development
Ikebe et al. *Biochemistry and Biophysical Research Communications*, 1998 [35]	LIMK2 mouse	adult liver, brain, thymus, lung, heart, stomach, testis	Northern blot, RT-PCR	Identification of LIMK2c, a brain-specific isoform, and LIMK2t, a testis-specific isoform
Takahashi et al. *Biochemistry and Biophysical Research Communications*, 1998 [47]	LIMK1 LIMK2 mouse	adult brain, thymus, lung, spleen, testis, kidney, stomach, heart	Northern blot, in situ hybridization	LIMK2 expressed in all tissues, identification of a testis-specific isoform LIMK2t
Nomoto et al. *Genes*, 1999 [37]	LIMK2 human	fetal and adult brain, stomach, colon, pancreas, liver, lung, kidney, placenta	RT-PCR, RNase protection assay	Identification of LIMK2a and LIMK2b with tissue-specific expression profile. LIMK2a predominantly expressed in fetal and adult tissues compared to LIMK2b
Meng et al. *Neuron*, 2002 [48]	LIMK1 mouse	adult brain	LIMK1 KO mice, immunohistochemistry, primary neurons, brain sections	Dendritic spine morphology and synaptic function alterations
Takahashi et al. *Developmental Biology*, 2002 [41]	LIMK2 mouse	adult testis	LIMK2 KO mice, MEF cells, immunofluorescence, histology	Abnormal spermatogenesis found in LIMK2-KO testis associated with an increased number of apoptotic germ cells
Meng et al. *Neuropharmacology*, 2004 [49]	LIMK1 LIMK2 mouse	adult brain	LIMK1, LIMK2 and LIMK1/2 KO mice, immunohistochemistry, electrophysiology	Normal synaptic plasticity in LIMK2-KO mice, altered synaptic functions in double-LIMK1/2-KO mice
Chen et al. *Current Biology*, 2004 [50]	dLIMK drosophila	from larvae to adult leg morphogenesis	mRNA level	Expression in late larval and pupal stages, suggesting a role in this transition. Defects in leg morphogenesis. Role of the Rho-dLIMK signaling pathway.
Foletta et al. *Experimental Cell Research*, 2004 [51]	LIMK1 rat, mouse, chicken	rat and chick embryos brain and spinal cord mouse adult brain, heart, liver, lung, small intestine, stomach, kidney	Western blot	Expression of LIMK1 in liver, thymus, kidney, heart, lung, small intestine, stomach and brain
Ang et al. *Developmental Biology*, 2006 [52]	dLIMK drosophila	larvae neuromuscular junctions (abdominal muscle fibers), antennal lobe glomeruli	dLIMK active/inactive, drosophila strains, immunohistochemistry, electrophysiology	Role of LIMK in synapse development and in glomeruli of antennal lobe. LIMK is a downstream effector of PAK
Acevedo et al. *Journal of Histochemistry and Cytochemistry*, 2006 [38]	LIMK2 mouse	E14 olfactory epithelium, heart, liver, intestine, urogenital sinus, thymus, spinal cord adult brain, heart, spleen, stomach, intestine, lung, skin, kidney, ovary, eyes, testes, uterus	Embryo sections, immunohistochemistry, western blot	Variable LIMK2 expression levels in embryonic and adult tissues, similar expression pattern than LIMK1 except in testis
Menzel et al. *Mechanism of Development*, 2007 [53]	dLIMK drosophila	adult eyes	Genetic screen, mutant, drosophila strains, immunohistochemistry of photoreceptor cell	PAK-LIMK-cofilin pathway are involved in photoreceptor cell morphogenesis by regulating adherent junctions and actin dynamics
Ott et al. *Gene Expression Patterns*, 2007 [54]	LIMK1 LIMK2 zebrafish	all embryonic stages	In situ hybridization	Temporal and spatial expression of LIMK1 and LIMK2 during embryogenesis
Lindström et al. *Gene Expression Patterns*, 2011 [33]	LIMK1 mouse	E10.5 to E18.5 EMT- and MET-tissues, limb, eye, heart, lung, skin, kidney, intestine, testes	Embryo sections, immunohistochemistry	LIMK1 highly expressed in many neuronal and epithelial tissues undergoing EMT and MET
Rice et al. PLoS ONE, 2012 [40]	LIMK2 mouse	E14.5 E15.5 E18.5 and newborns P1.5 ocular tissue adult brain, testis, eyes, rate, lung	LIMK2-KO mice, RT-PCR, in situ hybridization, western blot, immunohistochemistry	Phenotype of EOB “eyes open at birth” of LIMK2-KO mice, abnormal migration of keratinocytes during eyelid development
Andrews et al. *Biology Open*, 2013 [55]	LIMK2 mouse	E13.5, E15.5 brain	In situ hybridization, siRNA transfections, in utero electroporation	Role of LIMK2 in growth cone collapse in response to Sema3A by regulating PlexinA1 expression level
Kawano et al. *Bone*, 2013 [56]	LIMK1 mouse	newborns PD3-PD5 tibiae, femur	LIMK1-KO mice, bone histomorphometry, microCT, primary osteoblasts, osteoclasts and bone marrow cells	Bone mass reduction in LIMK1-KO mice, abnormal osteoblast differentiation and defective osteoblastic and osteoclastic functions
Abe et al. *Development*, 2014 [57]	dLIMK drosophila	newborns P2-P3 brain	Drosophila strains, immunohistochemistry	Involvement of Rac-Sickie-SSH and Rac-PAK-LIMK pathways in axonal growth
Piccioli et al. *Journal of Neuroscience*, 2014 [58]	dLIMK drosophila	larvae neuromuscular junctions	Drosophila strains, live imaging of synaptic growth and bouton budding	Role of BMPRII-LIMK-cofilin-actin signaling in potentialization of neuromuscular junctions
Yang et al. *Bone*, 2015 [59]	LIMK2 mouse	newborns PD2-PD3 primary osteoblasts	siRNA transfections, immunofluorescence, fluid shear stress	Contribution of LIMK2 in the mechanosensitivity of osteoblasts
Xie et al. *Histochemistry and Cell Biology*, 2017 [60]	LIMK1 mouse	E15.5, E18.5, newborns P1 brain	In utero electroporation, brain sections, immunofluorescence	Altered neuronal migration and number of neurites due to aberrant expression of LIMK1
Duan et al. *Cell Cycle*, 2018 [32]	LIMK1 LIMK2 mouse	2, 4, 8 -cells morula, blastocyst	In vitro fertilization, embryo culture, immunofluorescence, inhibition of LIMKs activity	LIMK1 and LIMK2 are involved in early stages of embryo development and regulate actin assembly
Saxena et al. *Development*, 2018 [61]	LIMK mouse	E11.5, E13.5, E15.5 cortex and cortical neurons postnatal P0, P6, P21 neurons	Immunohistochemistry, cell proliferation, in utero-electroporation, P-SMAD labeling	Regulation of dendritic branching by LIMK-mediated non-canonical BMP signaling and involvement of both canonical and non-canonical BMP signaling in neuronal migration
Mao et al. *Molecular Brain*, 2019 [62]	LIMK1 LIMK2 mouse	E14.5 brain	LIMK1-KO, LIMK2-KO and double LIMK1/2-KO mice, immunohistochemistry	Contribution of LIMK1 and LIMK2 in progenitor cell proliferation and migration. Role of LIMK2 in embryonic cell apoptosis
Fang et al. *Scientific Reports*, 2019 [63]	LIMK1 LIMK2 mouse	E3 and from P3 to P30 cochlea	LIMK1/2-KO mice, immunohistochemistry, auditory measurement	No alteration of cochlear development and auditory function in LIMK1/2-KO mice
Kwon et al. *Asia-Australasian Journal of Animal Sciences*, 2020 [31]	LIMK1 LIMK2 porcine	1, 2, 4-cells morula, blastocyst	RT-QPCR, LIMK1/2 activity inhibition, LIMK1/2 dsRNA injection, embryo culture, immunofluorescence	Role of LIMK1 and LIMK2 in embryo cleavage and compaction through actin regulation and the maintenance of cell–cell junctions
He et al. *In Vitro Cellular and Developmental Biology*, 2021 [64]	LIMK2 human	Embryonic Stem Cells	endodermal differentiation, siRNA transfection, RT-QPCR, immuno-fluorescence	Control of actin assembly, EMT-related genes expression and cell migration by LIMK2 in endodermal lineage

## 3. Signaling Pathways Involving LIMKs during Development

While LIMKs are actors of several signaling pathways, the best-described pathways during development are the semaphorin, nerve growth factor and the non-canonical bone morphogenetic protein (BMP) pathways. Within these pathways, LIMKs are downstream effectors of RhoA/ROCK, cdc42/PAK, and Rac/PAK (for reviews, see [12,21,22]). 

### 3.1. The Non-Canonical BMP Pathway

BMPs belong to the TGFβ superfamily and play a crucial role during the development of the nervous system, where LIMK1 is abundantly expressed. Thus, the BMPs/LIMK1/cofilin axis has been thoroughly investigated and has been shown to be essential for neuronal morphogenesis. 

BMPs bind to a heterodimeric complex of type-I and type-II Ser/Thr kinase receptors and, as a result, type-II receptors phosphorylate and thereby activate type-I receptors. While BMP can signal through Smad-dependent (canonical) and Smad-independent (non-canonical) pathways, LIMK1 is only involved in the non-canonical Smad-independent pathway. LIMK1 interacts with a large cytoplasmic domain of 600 amino acids of BMP type-II receptors (BMPRII) [65,66]. According to Foletta and collaborators, this interaction prevents the activation of LIMK1 by PAK [65]. This downregulation of LIMK1 can be relieved by the binding of BMP4 to its receptor, leading to the dissociation of LIMK1 from BMPRII and subsequently to cofilin phosphorylation (Figure 1).

Lee-Hoeflich and collaborators demonstrated that BMPs induce the activation of the small GTPase cdc42, which cooperates with the binding of LIMK1 to BMPRII to prompt high levels of LIMK1 activity and of phospho-cofilin [66]. This may be a useful mechanism to control cofilin phosphorylation close to the plasma membrane, to locally regulate cortical actin dynamics.

The activation of LIMK1 by BMPs has been further characterized during dendritogenesis, by experiments showing that PAK1 binds to BMPRI (BMPR1b and ALK2), resulting in the close proximity of the different players in this signaling pathway [61,67,68] (Figure 1).

In order to establish a correct neuronal circuit, the growth rate and length of neurites must be controlled in space and time during the different stages of neuronal development. This implies a fine regulation of actin dynamics by BMP/LIMK1/cofilin signaling, as illustrated by the work of Wen and collaborators, who established that BMP gradients are required to trigger the rapid responses of neuronal growth cones via LIMK1- and SSH-regulated cofilin phosphorylation [69]. The BMP gradients act as guidance cues to orient the commissural axons in the developing spinal cord [70]. Commissural axons extend away from the roof plate in response to BMP chemorepellent activity, which controls the growth rate of commissural axons and their orientation by finely regulating LIMK1 activity. When the BMP/LIMK1/cofilin signaling pathway is activated, the growth rate is slowed down, whereas when LIMK1 activity is reduced, axons grow faster [70]. Thus, temporal and spatial guidance decisions result from the activation status of cofilin and actin dynamics in a LIMK1-dependent manner during development [68,70]. Frendo and collaborators, using a mouse LIMK1-KO model, showed that such a regulation of cofilin activity and actin dynamics by LIMK1 is relevant in axon regeneration after sciatic nerve injury [71]. 

Taken together, these studies highlight the fine modulation of LIMK1 activity by BMPs to control actin dynamics during neuronal development. 

Although LIMK1 appears to be a preferred partner of BMPRII in the developing neurons, LIMK2 is also activated by BMPs. Its direct interaction with BMPRII was, however, debated between Foletta and collaborators [65] and Lee-Hoeflich and collaborators [66], but was demonstrated in the context of cancer development [72]. The BMP-induced activation of LIMK1 and LIMK2 occurs via two distinct signaling pathways that are both independent of SMAD. BMP-induced LIMK1 activation involves cdc42 and PAK signaling, whereas BMP-induced LIMK2 activation occurs via RhoA and ROCK signaling. Other factors, such as members of the TGF family (TGFbeta1 and Activin B), also activate LIMK2, via RhoA and ROCK, to control cofilin phosphorylation and actin dynamics [73,74].

### 3.2. Nerve Growth Factor

A critical role of both LIMK1 and LIMK2 in controlling the cofilin activity during nerve growth factor-induced neurite extension and growth cone motility was mentioned by Endo and collaborators [75]. Interestingly, this study also revealed that the respective roles of LIMK1 and LIMK2 are distinct and that the signaling mechanisms leading to their activation, as well as their activation time-course, differ.

### 3.3. Semaphorins

The semaphorins 3A and 3F (Sema3A/3F) are chemorepulsive axonal guidance molecules, acting through neuropilin-plexin receptors expressed at the surface of neurons. 

By using a dominant-negative LIMK1, which cannot be activated by PAK or ROCK, Aizawa and collaborators showed that LIMK1 activation is required for Sema3A-induced growth cone collapse in dorsal root ganglia neurons. In addition, a synthetic cell-permeable peptide containing a cofilin phosphorylation site acts as a competitive inhibitor of LIMK1, which leads to a reduction of cofilin phosphorylation and the suppression of Sema3A-induced growth cone collapse [76].

Further investigations by Duncan and collaborators provide evidence that Sema3F signaling in cortical neurons sets up a pathway downstream of Rac1, involving the phosphorylation of PAK1-3, LIMK1/2 and cofilin, which leads to spine retraction [77]. The role of LIMK2 was analyzed more precisely, in terms of the semaphorin response of cortical interneurons during embryonic development, by Andrews and collaborators. By combining in vitro and in vivo experiments of silencing LIMK2 with siRNA, the authors showed that interneurons deficient in LIMK2 are not responsive to Sema3A signaling. Furthermore, LIMK2 knockdown is concomitant with a reduction in the expression level of the receptor PlexinA1, a co-receptor involved with neuropilin in Sema3A binding. The molecular mechanism is not clearly elucidated but it appears that LIMK2 mediates the response to Sema3A via the control of the expression level of PlexinA1 [55]. This results in the abnormal migration of these interneurons and a higher number of neurites with reduced length [55]. These studies suggest that both LIMK1 and LIMK2 are involved in the semaphorin pathway, not only by acting on cofilin activity and actin dynamics but also via different partners of each kinase. LIMK2 seems to be specifically involved in the SEMA3A pathway, while LIMK1 is involved in SEMA3A and SEMA3F signaling.

## 4. The Role of LIMKs during Embryonic Cell Migration

Directed cell migration is an integrated process that is essential for embryonic development and takes place throughout the developmental processes, from the earliest stages, during gastrulation, and during important determination steps, leading to appropriate differentiation in space and time. Although numerous studies have suggested that LIMKs regulate cell motility through cofilin phosphorylation, evidence supportive of this function during migration processes occurring in vivo are still limited to neural progenitors and the keratinocytes of nascent eyelids.

### 4.1. Role of LIM Kinases in Neural Progenitor Migration

Although LIMKs are expressed in the central nervous system during development [36,51], the precise role of LIMK signaling in cortical development has been investigated only recently, using LIMK1 KO, LIMK2 KO or LIMK1/2 double-KO mice [62]. Whereas the overall laminar organization of the cortex was not altered in newborn mice, a reduced number of late-born pyramidal neurons was observed in all three KO mouse lines. Tracing the outcomes at the birth of progenitors labeled with BrdU on the embryonic day 14, the authors showed that mice lacking LIMK exhibited significant deficits in neural progenitor proliferation and migration. This study highlights the critical role of LIMKs in the proliferation and migration of neural progenitors, although the exact function of each LIMK in these processes and the stage of their action remain to be discovered. It can be proposed from the work of Das and Storey [78] and Kawaguchi [79] that LIMKs play a critical role during the first delamination prior to migration, as this mechanism is dependent on actin-myosin contraction and involves dynamic changes in actin and adherent junction organization. Indeed, during cortical development, neurons and differentiating neuronal cells, called intermediate progenitors, are generated by the division of neural progenitor cells from the neuro-epithelial layer. The newborn neuronal daughter cell has to escape the epithelial tissue, via a necessary delamination step that will lead to a retraction of the cellular processes that link the neuroepithelial cell to the apical surface. 

### 4.2. Role of LIM Kinases in Keratinocyte Migration

One specific role of LIMK2 in the control of keratinocyte migration has been demonstrated in LIMK2-deficient mice. Indeed, in the absence of LIMK2, keratinocytes in nascent eyelids differentiate and acquire a pre-migratory phenotype, but the leading keratinocytes, emerging from the tip of the eyelid, fail to nucleate filamentous actin and subsequently do not migrate [40].

This results in an eye-open at birth (EOB) phenotype that closely resembles the one observed in ROCK KO mice [80,81]. This indicates that LIMK2 is the biochemical target of ROCK that modulates actin dynamics during keratinocyte migration and is important for eyelid closure. Interestingly, the EOB phenotype has never been described in LIMK1-deficient mice (personal observations and [56,82]), which suggests that each LIMK is regulated differently, to perform different functions.

Finally, this central role of LIMK2 in the migration of leading keratinocytes can be related to a previous work showing that, during cell invasion in vitro, LIMKs are required for path generation by leading the tumor cells and nontumor stromal cells during collective tumor cell invasion [83]. This similarity reinforces the concept that mechanisms at work during embryonic development are reactivated during the cancer process.

## 5. Role of LIMKs in Epithelial-Mesenchymal Transitions

The epithelial–mesenchymal transition (EMT), a physiological process by which epithelial cells lose their polarity and cell–cell adhesion and gain migratory and invasive properties to become mesenchymal cells, is at work during specific developmental steps. EMT is also observed in cancer, as a process that may also favor cell-invasive properties and promote carcinoma progression [84,85].

The first EMT events taking place in the embryo, referred to as primary EMT, are associated with major induction events, giving rise, for example, to mesoderm formation during gastrulation or to neural crest cell delamination from the neural tube [84]. It is difficult to establish indications regarding EMT events during the gastrulation step in vivo, as major perturbation during this early event is usually lethal or is compensated for by the upregulation of other signaling pathways. 

Interestingly, an EMT accompanies the derivation of endoderm cells from human pluripotent stem cells (hPSCs) in vitro and, therefore, allows researchers to mimic and finely study the gastrulation events taking place during early vertebrate development [64]. In fact, knocking down LIMK2 by siRNA is sufficient to inhibit the EMT process during endodermal lineage specification. This inhibition was accompanied by the absence of upregulation of EMT-associated genes, such as SNAIL 1 and 2 transcription factors. Cell migration was also impaired in the absence of LIMK2 in these induced hPSCs. This work emphasizes one particular role of LIMK2 in endodermal lineage specification, via the regulation of key EMT transcription factors through actin cytoskeletal assembly [64]. In the same way, experiments based on a kinome-wide RNAi screen, to identify kinases that regulate somatic cell reprogramming to iPSCs, have shown that the knockdown of LIMK2 or the cofilin kinase TESK1 in mouse embryonic fibroblasts (MEFs) is an inducer of mesenchymal to epithelial (MET) transition, the reverse process of EMT. Indeed, in the absence of LIMK2 or TESK1 during iPSCs reprogramming, cofilin phosphorylation is decreased and the actin cytoskeleton is disrupted [86]. Although still difficult to reconcile, these results point to a central role of LIMK2 activity in the control of the transition between the epithelial and the mesenchymal phenotypes.

Neural crest cells (NCC) are multipotent cells at the border of the neural plate, which acquire a migration phenotype at the time of neural tube closure and will later give rise to diverse cell lineages, contributing to most of the peripheral nervous system, the craniofacial cartilage and bones, as well as pigment cells [87,88]. Neural crest cells undergoing EMT have been intensively studied to understand the mechanism of delamination. Chick embryos provide a useful in vivo model to follow the first step of NCC de-epithelialization. By manipulating LIMK1 expression in chicken embryos, through in ovo electroporation in the neural tube prior to NCC delamination, Park and Gumbiner have shown that LIMK1 overexpression induces de-epithelialization in the neural tube, whereas a dominant-negative LIMK1 inhibits de-epithelialization [89]. More indirectly, in the same model of chick embryo neural crest cells, matrix metalloproteinase 14 (MMP14) has been shown to be required for NCC delamination [90,91]. Interestingly, MMP14 is a recently identified substrate of LIMK1/2, and its phosphorylation has been shown to play a role in the endosome-mediated recycling of MMP14 to invadopodia and matrix degradation in MDA-MB-231 breast carcinoma cells, thereby contributing to the functional machinery required for invasion [92].

During the organogenesis processes that occur later, when tertiary EMT takes place [84], LIMK1 is highly expressed in the organs where EMT or MET takes place [33]. 

Additionally, LIMK1 has been shown to co-localize via immunochemistry with Wilms’ tumor protein 1 [93], which regulates EMT in the epicardium via the activation of SNAIL and the inhibition of E-cadherin expression.

Together, these results highlight an important role for LIMKs in the control of transitions between epithelial and mesenchymal phenotypes during embryo development, with possibly a preponderant involvement of LIMK2 in the early stages, while LIMK1 may act later during development.

## 6. Impact of LIMKs in Cell Differentiation

Besides the differentiation of cells in epithelial and mesenchymal phenotypes, LIMKs participate in other differentiation processes that occur during development or in the adult, such as differentiation of neurons, bone cells, gametes, blood cells or gland morphogenesis.

### 6.1. Neuron Differentiation 

The only LIM kinase present in Drosophila melanogaster was found to be primarily involved in the development of olfactory and neuromuscular synapses [52], highlighting the importance of this enzyme in neuronal differentiation.

One specific role for LIMK1 has been described in the control of dendritic spine morphogenesis. Dendritic spines in the mammalian central nervous system are small, specialized post-synaptic protrusions on the dendrites where excitatory chemical synapses are formed. Actin dynamics are involved in the formation, morphological properties, and motility of the dendritic spines, as well as in presynaptic neurotransmitter release, post-synaptic receptor function and synaptic plasticity [94].

The first evidence of the involvement of LIMKs in synaptic function was provided by Meng and collaborators, who have shown that LIMK1 knockout mice exhibited normal anatomy of the brain but significant abnormalities in spine morphology and in synaptic function, including enhanced hippocampal long-term potentiation (LTP), a persistent increase in synaptic strength following high-frequency stimulation [48]. The LIMK1 knockout mice showed altered fear responses and spatial learning, which are reminiscent of some of the symptoms of Williams–Beuren syndrome. LIMK2 knockout mice, on the other hand, exhibited only minimal LTP abnormalities [49]. 

In 2015, George and collaborators have further demonstrated that a conserved palmitoyl-motif is necessary and sufficient to target and anchor LIMK1 in the spine. Using shRNA knockdown, followed by rescue experiments, they revealed that LIMK1 palmitoylation is essential for normal actin polymerization in spines, for spine-specific structural plasticity, and for long-term spine stability [95]. Recently, a study performed by Chen and collaborators has also shown that neuregulin, a transmembrane ligand of the plasma membrane kinase receptor ErbB4, interacts with the LIM domain of LIMK1 to regulate its activity, which affects spine density [96]. In addition, mutations or altered expression levels of neuregulin are linked to increased susceptibility to schizophrenia and depression. Conversely, pharmacological LIMK inhibition restores normal behavior in a mouse model of schizophrenia [97]. 

### 6.2. Bone Cell Differentiation

In addition to its role in neurogenesis, LIMK1 is required for proper bone formation. LIMK1 is highly expressed in bone at equivalent levels in osteoclasts and osteoblasts, the two cell types on which bone homeostasis depends. Osteoclasts resorb the bone matrix and osteoblasts secrete the new bone matrix, allowing continuous renewal of the skeleton throughout life. By examination of the skeletal phenotype of LIMK1-KO mice, Kawano and collaborators observed a reduced bone mass at different skeletal sites [56]. Histomorphometric analysis of LIMK1-KO bones revealed a significant decrease in osteoblast numbers, whereas osteoclast numbers were normal. This reduced number of osteoblasts is due to a low number of osteoblast progenitors, in LIMK1-KO bone marrow. This is explained by an impaired osteoblastic differentiation of stromal stem cells and a decrease in cell viability in the absence of LIMK1 [98]. LIMK1-KO osteoblasts are functionally abnormal, exhibiting defective mineralizing capacity. In addition, the absence of LIMK1 alters the osteoclast function in bone, with an observed increase of their bone resorptive capacity. LIMK1-KO osteoclasts also showed a greater spreading response to colony-stimulating factor 1. Increased spreading is explained by an enhanced actin treadmilling in lamellipodia, due to more active cofilin in the absence of LIMK1 [56]. Thus, the loss of LIMK1 activity results in dramatically altered osteoblast and osteoclast functions, and osteoblast differentiation.

In contrast, the role of LIMK2 in bone homeostasis is less widely investigated. To date, no in vivo studies have been performed on bone development in LIMK2-KO mice. Only one in vitro study has been published, using normal primary mouse osteoblasts in which the LIMK2 gene was silenced by siRNA [59]. In these experiments, Yang and collaborators proposed that osteoblast mechanosensitivity and cell proliferation are increased when LIMK2-deficient cells are under continuous mechanical stress [59]. Thus, by controlling stress fiber formation in osteoblasts, LIMK2 regulates the osteoblast’s mechanosensitivity and contributes to bone mass formation and osteogenesis. 

Taken together, these data revealed that both LIMKs have central but distinct roles in bone formation.

### 6.3. Gonadal Cell Differentiation

Spermatogenesis and oocyte maturation offer other examples of differentiation processes in adults, wherein LIMKs play a regulatory role during important actin reorganizations.

Sperm differentiation encompasses a complex sequence of morphological changes that takes place in the seminiferous epithelium. A testis-specific isoform of LIMK2 (tLIMK2) that lacks LIM domains and part of the PDZ domain is specifically expressed in differentiated, meiotic stages of spermatogenic cells [47]. Moreover, impaired spermatogenesis and germ cell loss, in association with enhanced apoptosis in spermatocytes, have been observed in LIMK2-deficient mice, with disruption of the three LIMK2 isoforms [41]. Interestingly, LIMK2 ranks high in the comprehensive list of 654 human infertility-related genes that have been sequenced in the study of nonobstructive azoospermia, the most common form of azoospermia, performed by Li and collaborators [99]. This central role in spermatogenesis is specific for LIMK2, as it has not been observed in LIMK1-deficient mice and is not reported in the study by Li and collaborators. In contrast, LIMK1 is essential for the ability of spermatozoa to fertilize. In fact, in mice, it has been shown that LIMK1 activity is central to the correct execution of acrosomal exocytosis during the capacitation process [100]. 

Mammalian oocyte maturation is distinguished by asymmetric division and polar body extrusion, which is regulated primarily by the cytoskeleton, including microtubules and microfilaments. The first evidence of a functional role of LIMKs in oocyte maturation was obtained in Xenopus. It was observed that the mRNAs encoding the Xenopus homologs of mammalian LIMKs, XLIMK1 and XLIMK2, are abundantly expressed in oocytes. Microinjection experiments placing XLIMK1/2 mRNA into progesterone-treated oocytes prevented Xenopus cofilin dephosphorylation (activation) and entry into meiosis. This is countered by the co-injection of XLIMK1/2 with constitutively active cofilin [101]. Based on these results, it was proposed that XLIMK is a regulator of cytoskeletal rearrangements during oocyte maturation.

LIMK inhibition through the application of the LIMK inhibitor LIMKi3, or microinjection of a LIMK1 antibody into mouse oocytes, induces a defect in the positioning of the microtubule-organizing center [102] and of the meiotic spindle positioning [103]. Moreover, it has recently been shown that disruption of LIMK1/2 activity, either through LIMK1- and LIMK2-specific siRNA microinjection or LIMKi 3 inhibitor treatment, significantly decreases oocyte polar body extrusion and induces abnormal spindles. The small GTPase RhoA is the upstream regulator of LIMK1/2 action on actin assembly and spindle organization [104]. 

The role of cofilin and its regulators in the control of actin network dynamics, during the different stages of mouse oocyte maturation, was explored in depth in a recent study. It was found that cofilin activity varies depending on the stages of meiosis, with an inactivation through its phosphorylation during prophase arrest. Successful maturation, with the resumption of meiosis, involves a switch-like activation of cofilin [105]. 

### 6.4. Gland Morphogenesis

Branching morphogenesis is a developmental process used by the developing organs, such as mammary and salivary glands. Using small interfering RNA (siRNA) knockdown or a small-molecule LIMK inhibitor (LIMKi3), the function of LIMK 1 and 2 and their downstream effectors in this morphogenetic process was investigated using organotypic ex vivo cultures of embryonic mouse submandibular salivary glands. This study identified a central role for LIMK 1/2, coordinating the actin and microtubule cytoskeleton in the process of cleft formation during salivary gland branching morphogenesis [106]. 

Similarly, it has been demonstrated that a Cdk5-Pak1-LIMK-Cofilin kinase cascade regulates intrahepatic biliary network branching in zebrafish [107]. 

### 6.5. Blood Cells 

Many of the processes observed during embryonic development also take place throughout adulthood, as illustrated by the differentiation of sexual cells described above, or, for instance, the renewal of polarized cells in the epithelium of the gut, the endothelial cell differentiation during angiogenesis in the context of wound healing, or the daily renewal of hematopoietic cells. 

The aberrant differentiation of megakaryocytes in the absence of the upstream LIMK regulators RhoA and Cdc42 is an illustration of the importance of LIMKs to differentiation processes in the adult organism. Megakaryocytes are the progenitors of blood platelets and undergo a remarkable differentiation process that involves endomitoses and the maturation of the cytoplasm, to allow the formation of long cytoplasmic processes called proplatelets [108]. In fact, conditional double knock-out mice for the two small GTPases, RhoA and Cdc42, have megakaryocytes with increased LIMK and cofilin expression levels. These megakaryocytes are still able to undergo endomitoses but the normal maturation process of their cytoplasm does not take place, which results in a macrothrombocytopenia phenotype in these mice [109]. This observation reinforces previous studies in which phosphorylated, inactive cofilin or cofilin deficiency correlates with altered proplatelet formation and macrothrombocytopenia [110,111]. Interestingly, in two studies focusing on megakaryocyte maturation, reduced LIMK expression and cofilin phosphorylation are also accompanied by an abnormal actin cytoskeleton and less proplatelet formation, suggesting that the right balance of activities might be essential for efficient megakaryocyte differentiation [112,113]. 

Mast cell differentiation is also under the control of a signaling pathway that includes LIMKs since their deficiency reduces the proliferation and maturation of bone marrow-derived mast cells [114]. Similarly, LIMKs are part of the signaling cascade responsible for monocyte to macrophage differentiation [115]. 

Besides differentiation processes as such, LIMKs also play a key role in the activation processes of certain cell types, necessitating the reorganization of the cytoskeleton. For instance, the activation of blood platelets is initiated by the coiling of marginal band microtubules, as well as by actomyosin contraction; this is followed by spreading and the formation of actin stress fibers [116]. It has been shown that LIMK1 and the LIMK2a isoforms are expressed in platelets, and their presence/activity is essential for the platelet activation and spreading process [117,118]. Furthermore, there appears to be an additional function of LIMK1 in platelets that is independent of its role in actin polymerization, because LIMK1 deficiency leads to reduced thromboxane production following von Willebrand factor-induced platelet activation, which is independent of the actin polymerization status [119]. 

Another example is the activation of microglia in the nervous system that undergo morphological changes induced by a signaling cascade, including LIMK/cofilin actions [120]. LIMK expression is also essential for natural killer cell cytotoxicity by ensuring lytic granule trafficking to the immune synapse [121]. 

## 7. Future Directions

Given their central role in the regulation of actin dynamics, one would expect that the KO of LIMKs would be lethal. This is not the case, suggesting that compensatory mechanisms may act to counteract LIMK deficiency. Furthermore, only the most obvious KO phenotypes were identified, such as in problems with neuronal morphogenesis or bone cell differentiation in LIMK1-deficient mice. More subtle disturbances in KO mice could possibly be revealed by challenging them with particular environmental situations. This could allow the identification of other implications of LIMKs and probably of specific functions for each isoform. 

Although it has long been known that LIMK1 and LIMK2 have different tissue distributions, it is still not clear what could be the purpose for higher eukaryotes to have two paralogous forms of LIMKs. Selective inhibitors of LIMK1 have recently been described [122]. Their use, as well as the development of selective LIMK2 inhibitors, may provide some answers to this question [30]. Furthermore, although cofilin is the best-described LIMK substrate, other substrates are beginning to be discovered, extending our knowledge about the field of action of LIMKs. The possibility is not excluded that specific substrates for each of the LIMKs will one day be discovered, unraveling their specific role.

Moreover, the study of the role of the LIMK’s PDZ and LIM modules in regulating LIMK activity, subcellular localization, the formation of complexes and substrate phosphorylation remains a poorly explored, yet exciting, field.

The increasing number of publications describing a role for LIMKs at almost all stages of development and in adults suggests an important role of LIMKs also in pathological situations. Indeed there are numerous reviews that report deregulation of the cofilin pathway, including LIMK activity/expression in neurological and mental disorders [25,29,123], in cancer and metastasis [28,124], and in several urogenital diseases such as erectile dysfunction and infertility [125].

The detailed study of the physiological processes in which both LIMKs are involved may give us some clues on diagnostic and therapeutic approaches, in order to restore the correct level of their activities when they are disturbed in pathological situations. 

## Figures and Tables

**Figure 1 cells-11-00403-f001:**
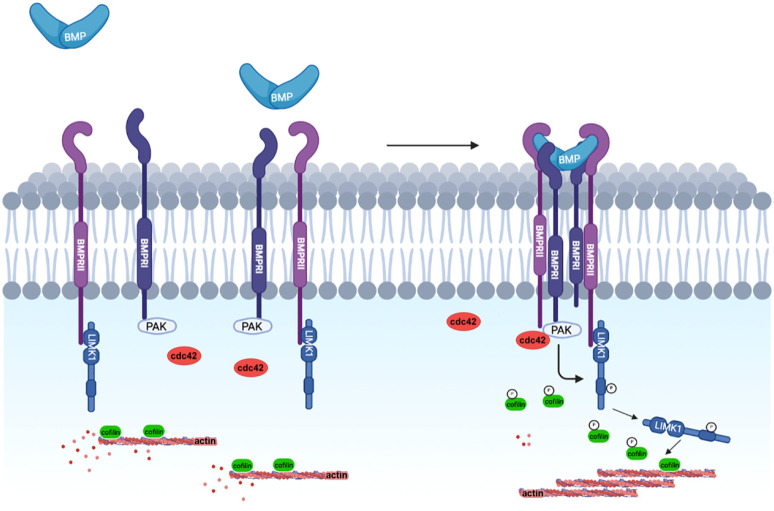
The binding of BMPs leads to the formation of a BMP receptor complex. PAK1, recruited by BMPRI, is thus in close proximity to its target LIMK1, bound to BMPRII. After being activated by cdc42, PAK1 phosphorylates LIMK1, which induces cofilin phosphorylation, thereby inhibiting its activity and thus contributing to the regulation of actin dynamics required for dendrite extension.

## Data Availability

Not applicable.

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
