# Peer review of "The Role of LIM Kinases during Development: A Lens to Get a Glimpse of Their Implication in Pathologies"

_cells, 2022, doi:10.3390/cells11030403_

Round 1

Reviewer 1 Report

In this review, the authors describe the functions of the LIMK family on the development of various tissues and cells. Although many reports have been published on the functions of LIM-kinases, authors chose and cited the essential reports from at the time of discovery of LIM-kinases to recent. The contents are consistent with the cited reports, and the manuscript is well organized. There are only a few minor mistakes. I can't find anything that needs to be revised.

Minor points:

line 67-75: The authors did not cite the report by Frangiskakis et al., Vol. 86, p59–69, Cell (1996) on the relationship between Williams syndrome and the LIMK1 gene. The authors should cite the report and briefly describe the contents.

Line 281-282, ref 84: The authors are better to add the name of TESK, a cofilin kinase, to this content.

Line 300: Ref 87 (Steffen et al.) should be with Ref 86 on line 296.

 Typographical error

line 250:  "observations),"

 Line 302: (Thiery et al. 2009) -> Ref 83 ?

Author Response

Dear Editor,

Please find attached the revised version of our manuscript.

We thank the reviewers for their constructive comments and we have revised our manuscript accordingly. We have addressed all of the reviewer's comments and the resulting text modifications are highlighted in yellow.

Below is our detailed response to the reviewer’s comments, which have been copied, pasted and italicized for clarity.

We hope that you find this revised manuscript acceptable for publication in Cells

Sincerely yours,

Laurence Lafanechère, PhD

Reviewer #1

In this review, the authors describe the functions of the LIMK family on the development of various tissues and cells. Although many reports have been published on the functions of LIM-kinases, authors chose and cited the essential reports from at the time of discovery of LIM-kinases to recent. The contents are consistent with the cited reports, and the manuscript is well organized. There are only a few minor mistakes. I can't find anything that needs to be revised.

Minor points:

line 67-75: The authors did not cite the report by Frangiskakis et al., Vol. 86, p59–69, Cell (1996) on the relationship between Williams syndrome and the LIMK1 gene. The authors should cite the report and briefly describe the contents.

We thank the reviewer for this suggestion. We have added the reference (#26) and described the contents accordingly (Lines74-75)

Line 281-282, ref 84: The authors are better to add the name of TESK, a cofilin kinase, to this content.

Yes, we have added TESK (Lines 286-288)

Line 300: Ref 87 (Steffen et al.) should be with Ref 86 on line 296.

We have now correctly placed this reference (now #94), line 304, close to reference #93.

 Typographical error

line 250:  "observations),"

corrected

 Line 302: (Thiery et al. 2009) -> Ref 83 ? We have made the correction

Reviewer 2 Report

In the present review, authors have shared information on the roles LIMK1 and 2 during embryonic development. I have several reservations, my comments are appended below:

  1. Share the details of the experimental system referred to in references 1 and 2.
  2. The cellular migration and developmental stages are closely connected with the disease biology of cancer. Do authors find any disease conditions associated with LIMK1 and cofilin.
  3. The authors should present a table on the disease conditions associated with LIMK1.
  4. LIMKi3 inhibitor: share more details, how specific is it?

5. Authors should share the details on upstream activators for LIMK1.

  1. While referring to the earlier reports on LIMK1, the authors should mention the experimental system used.
  2. Role of LIMKs during embryonic cell migration and Role of LIMKs in epithelial-mesenchymal transitions: share representative figures.

8. Authors should make sure to justify every line with concerning reference, for instance, lines 258-261.

9. Reference 54- describe the model system used. The explanation given at this point looks vague.

  1. As authors note that LIMK1 impacts the dendritic spines, do authors find details on neurodegenerative disorders and LIMK1? I am aware that disease biology is not the main focus, a brief information may be okay.

11. There should be a ‘future directions’ section.

Author Response

Manuscript ID: cells-1543256

Dear Editor,

Please find attached the revised version of our manuscript.

We thank the reviewers for their constructive comments and we have revised our manuscript accordingly. We have addressed all of the reviewer's comments and the resulting text modifications are highlighted in yellow.

Below is our detailed response to the reviewer’s comments, which have been copied, pasted and italicized for clarity.

We hope that you find this revised manuscript acceptable for publication in Cells

Sincerely yours,

Laurence Lafanechère, PhD

Reviewer #2

In the present review, authors have shared information on the roles LIMK1 and 2 during embryonic development. I have several reservations, my comments are appended below:

  1. Share the details of the experimental system referred to in references 1 and 2.

The experimental system of reference 1 was described in the Table1. We have now added reference 2 in the table with the description of the experimental system used.

  1. The cellular migration and developmental stages are closely connected with the disease biology of cancer. Do authors find any disease conditions associated with LIMK1 and cofilin.

Deregulations of LIMK activity and of cofilin phosphorylation in cancer have been reported in several reviews. As the link between LIMKs and cancer is not the main scope of our present review, we only mention similarities regarding LIMK involvement in developmental and cancer processes (line 182, 263 , 268). Moreover, in the “future directions” paragraph, that we have added following the reviewer's suggestion,  we have now inserted the references of these reviews.

  1. The authors should present a table on the disease conditions associated with LIMK1.

As stated above, it seems to us that this would have extended the scope of our review too far. In the last section, we now refer to very complete reviews on the link between LIMKs and various pathologies.

  1. LIMKi3 inhibitor: share more details, how specific is it?

In line 93, we have added a short sentence about LIMKi selectivity and shared a recent reference about LIMK inhibitors (#30) which is published in the same series as this  submitted review.

  1. Authors should share the details on upstream activators for LIMK1.

In order not to disperse the reader, we have chosen to focus on signaling pathways that are especially involved in developmental processes and in which LIMKs are implicated. We do not provide a detailed description of the direct upstream activators but we cite the existing reviews about them (#12, 21, 22) in line 125-126.

  1. While referring to the earlier reports on LIMK1, the authors should mention the experimental system used.

Many of the experimental systems are described in Table 1. In addition, we have mentioned most of the experimental systems also in the text.

  1. Role of LIMKs during embryonic cell migration and Role of LIMKs in epithelial-mesenchymal transitions: share representative figures.

We do not quite understand the reviewer's request: does he want us to share figures already published in other journals on EMT or migration? We think this might be confusing in the context of our own review.

  1. Authors should make sure to justify every line with concerning reference, for instance, lines 258-261.

We thank the reviewer for this suggestion.  Indeed, we have gone through the whole manuscript and added the missing references. They are highlighted in yellow.

  1. Reference 54- describe the model system used. The explanation given at this point looks vague.

We have revised and completed this paragraph (lines 203-218), according to the reviewer's suggestion.

  1. As authors note that LIMK1 impacts the dendritic spines, do authors find details on neurodegenerative disorders and LIMK1? I am aware that disease biology is not the main focus, a brief information may be okay.

See our answer to point#3

  1. There should be a ‘future directions’ section.

In line with the reviewer's suggestion, we have added a section on future directions, replacing the former conclusion section.

Reviewer 3 Report

The review by Ribba and Colleagues deals with the role of LIM kinases during ontogenesis. It also tries to link the deficiencies in LIMK to human pathologies and to deficiencies seen in animal models of disease. The paper is very well written, and particularly useful is the table organized in chronological order, which allows to follow the development of knowledge within the field. 

Only a very minor note, 'new born' should read 'newborn' Lines 225 and 237).

Author Response

Manuscript ID: cells-1543256

Dear Editor,

Please find attached the revised version of our manuscript.

We thank the reviewers for their constructive comments and we have revised our manuscript accordingly. We have addressed all of the reviewer's comments and the resulting text modifications are highlighted in yellow.

Below is our detailed response to the reviewer’s comments, which have been copied, pasted and italicized for clarity.

We hope that you find this revised manuscript acceptable for publication in Cells

Sincerely yours,

Laurence Lafanechère, PhD

Reviewer #3

The review by Ribba and Colleagues deals with the role of LIM kinases during ontogenesis. It also tries to link the deficiencies in LIMK to human pathologies and to deficiencies seen in animal models of disease. The paper is very well written, and particularly useful is the table organized in chronological order, which allows to follow the development of knowledge within the field. 

Only a very minor note, 'new born' should read 'newborn' Lines 225 and 237).

We are pleased that the reviewer appreciated our work. We have corrected "newborn".

Round 2

Reviewer 2 Report

My suggestions are addressed